# Effects of returning corn straw and fermented corn straw to fields on the soil organic carbon pools and humus composition

Yifeng Zhang[a], Sen Dou[a,*], Batande Sinovuyo Ndzelu[a,b], Rui Ma[a], Dandan Zhang[a], Xiaowei Zhang[a,c], Shufen Ye[a], Hongrui Wang[a]

[a]College of Resource and Environmental Science, Jilin Agricultural University, Changchun, Jilin Province 130118, China
[b]National Engineering Laboratory for Improving Fertility of Arable Soils, Institute of Agricultural Resources and Regional Planning, Chinese Academy of Agricultural Sciences, Beijing, 100081, China
[c]College of Resource and Environmental Science, Nanjing Agricultural University, Nanjing, Jiangsu Province 210095, China

*Correspondence to*: Sen Dou (dousen1959@126.com)

**Abstract.** In our previous studies, we filtered out fungus (*Trichoderma reesei*) to have the best ability to transform corn straw into a humic acid-like substance through laboratory incubation experiments. In order to further verify our former findings, we set up a 360 day-field experiment that included three treatments applied under equal carbon (C) mass: (i) corn straw returned to the field (CS), (ii) fermented corn straw treated with *Trichoderma reesei* returned to the field (FCS-T), and (iii) blank control treatment (CK). Soil organic carbon (SOC), soil labile organic C components, soil humus composition, and the management levels of SOC pools under the three treatments were analyzed and compared. The results showed that the SOC content of CS and FCS-T treatments increased by 12.71 % and 18.81 %, respectively, compared with CK at 360 d. The humic acid carbon (HA-C) content of the FCS-T treatment was 0.77 g kg$^{-1}$ higher than in the CS treatment. Application of FCS-T appeared to promote the significant increase of SOC, carbon pool activity index and carbon pool management index through accumulation of HA-C, humin carbon, and easily oxidizable organic carbon (EOC) contents. Application of fermented corn straw treated with *Trichoderma reesei* (FCS-T) is more valuable and conducive to increasing soil easily oxidizable organic C (EOC) and humus C content than direct application of corn straw.

## 1 Introduction

Recycling and returning crop residues as soil amendments is an important prospect for increasing soil organic carbon (SOC) content and crop yield (Villamil et al., 2015), as well as managing crop straw residues. However, the decomposition process of crop residues is slow when they are directly applied to the soil (Zhang et al., 2019), and it is still not fully known how crop residues are transformed into stable SOC when applied to the soil (Cotrufo et al., 2013; Lehmann and Kleber, 2015; Zhang et al., 2015a). Information about decomposition and stability of carbon (C) is needed for long-term soil C sequestration (Cotrufo et al., 2013; Ndzelu et al., 2020a) and for reducing carbon dioxide (CO$_2$) emissions into the atmosphere (Chatterjee, 2013; Guan et al., 2018).

Contrary to direct crop residue application and conventional composting method, pre-treatment of crop residues with microbial inoculants is the most effective method for reusing crop residues as eco-friendly amendments to improve soil fertility and increase SOM (Bhattacharjya et al., 2021; Organo et al., 2022). This strategy accelerates crop residue degradation and humification (Vargas-Garcia et al., 2006; Ahmed et al., 2019; Nigussie et al., 2021; Sajid et al., 2022), by significantly halving the time needed for compost to reach maturity when compared to conventional composting methods (Organo et al., 2022). When applied to the soil, the microbial inoculant product improves C sequestration (Ahmed et al., 2019) and increase humic substances in the soil (Vargas-Garcia et al., 2006; Huang et al., 2008). The microbial inoculant-based fermentation produces contain partially degraded materials and organic compounds enriched of stable molecules (Huang et al., 2008). This fact, favours the accumulation and production of SOM containing both labile organic substances from the partially degraded portion and stable humic fraction.

Soil organic carbon is a good indicator of soil quality, but a suite of labile organic C components, such as water extractable organic carbon (WEOC), easily oxidizable organic carbon (EOC), and microbial biomass carbon (MBC) are effectively used to detect small changes in soil quality (Blair et al., 1995; Chen et al., 2009; Sainepo et al., 2018). This is because these labile organic C compounds are sensitive and promptly respond to changes in soil management practices (Blair et al., 1995; Xu et al., 2011), and they are also essential for the formation of the more stable SOC (Cotrufo et al., 2013). The labile SOC fractions are reported to be significantly affected by the application of organic amendments. Chen et al. (2017) and Ma et al. (2021) reported a significant increase in MBC and WEOC contents after crop straw residues were returned to the soil. In another study, Ndzelu et al. (2020b) also found that five years of corn straw application increased soil EOC, WEOC and MBC contents by 34.09%, 41.38% and 49.09% in the 0 − 20 cm depth, respectively. Therefore, assessing labile SOC fractions after crop straw applications may provide information about the formation of SOC (Chen et al., 2009; Huang et al., 2018; Liu et al., 2019; Ma et al., 2021). Another important index to monitor the effects of agricultural management practices on soil C sequestration is the carbon pool management index (Tang et al., 2018; Ma et al., 2021). The carbon pool management index, an index that includes SOC pools (carbon pool index) and SOC lability (carbon pool activity index), is widely used as a sensitive tool to determine changes in soil C content (Blair et al. 1995; Duval et al. 2019). A high carbon pool management index indicates that soil management practices have a greater potential to promote soil C sequestration (Duval et al. 2019).

Humic substance (HS) is the most stable fraction of SOM and contributes to the largest proportion to the total SOC (Olk et al., 2019; Dou et al., 2020). As a result, studying changes in soil humus components together with labile organic C fractions after corn straw application, could inform about the formation and stabilization of SOC during crop residues decay. Over the years extensive studies have been conducted to investigate the effects of crop residues on SOM and its pools (Atiyeh et al., 2002; Romero et al., 2007; Zhang et al., 2015a; Ng et al., 2016; Yang et al., 2020). Yet there are still conflicting reports and there is no general consensus about the effects of crop residue application on the formation and composition of SOM. For instance, recent studies have found that corn straw application significantly increases soil humus content and enrich soil humic acid structure with aromatic compounds (Fan et al., 2018; Zhang et al., 2019). While, other studies reported an

increase in aliphatic compounds in soils amended with corn straw (Yang et al., 2020; Ndzelu et al., 2020a). These diverging reports indicate that the magnitude and the influence of corn straw residues in SOM composition is unclear and site specific, warranting the need for more research to focus on the transformation of corn straw residues into SOM.

*Trichoderma*-mediated straw fermentation is gaining attention as a soil amendment and nutrient source (Gaind and Nain. 2006; Gaind and Nain. 2007; Siddiquee et al., 2017), and the role of *Trichoderma*-mediated straw fermentation in improving crop yield (Islam et al., 2014), promoting plant development, and alleviating biotic and abiotic stresses has been observed (Sarangi et al., 2021). In our previous studies (Yang et al., 2019; Zhang et al., 2020; Zhang et al., 2021), we observed in laboratory incubation experiments that the *Trichoderma reesei* (*T. reesei*) had the best ability to form humic acid-like during corn straw decomposition when compared with other fungi (*Phanerochaete chrysosporium* and *Trichoderma harzianum*). The use of *T. reesei* based fermentation has been employed in incubation studies and these studies show an increase soil C and humus composition (Gaind and Nain. 2007; Yang et al., 2019; Zhang et al., 2021). However, there is limited knowledge on the potential application of *T. reesei* fermented corn straw and increase SOM when incorporated into the field. In particular, the dynamic change process of different soil organic carbon components has not been reported yet. The objective of this study was to verify whether *T. reesei* can equally be effective in field trials to form relatively stable SOC fractions after corn straw application. We hypothesized that: (1) application of fermented corn straw treated with *T. reesei* (FCS-T) will be the most efficient in increasing soil humus content and soil C storage, due to the increase in aromatic C compounds; (2) application of FCS-T may also increase soil labile organic C components (WEOC, EOC, and MBC); and (3) application of FCS-T may also increase carbon pool management index level more than direct corn straw application. These assumptions are based on that *T. reesei* inoculant has strong humification ability compared with direct application of corn straw.

## 2 Materials and methods

### 2.1 Site description

A 360-day field experiment was conducted in a corn monocropping experimental field located at Jilin Agricultural University in Northeast China (N43°49′5″, E125°24′8″). Since 2005, monocropping of corn (*Zea mays* L.) has been the main cropping system in the region. The area is in a semi-humid region and receives a mean annual rainfall of 618 mm, with the highest precipitation occurring in the months of July and August. Soils in the study area are classified as Argiudolls according to the United States Department of Agricultural Soil Taxonomy (Soil Survey Staff. 2014). The basic soil characteristics are presented in **Table 1**.

### 2.2 Preparation and description of corn straw and fermentation of corn straw

Corn straw was collected from the adjacent cropland of corn (*Zea mays* L.) located at Jilin Agricultural University in Northeast China (N43°48′43.5″, E125°23′38.50″). The corn cultivar Zhongjin 368 type (Beijing Golden Grain Seed Co., Ltd.)

was planted at the end of April 2018 and harvested in early October 2018. After harvest, the whole corn straw residue was cut at the bottom and air-dried, thereafter shredded into 0.5 cm segments. A portion of the shredded corn straw was regarded as CS material.

The fermentation of corn straw was prepared by the fungal strains (*Trichoderma reesei* (*T. reesei*) MCG77) which were purchased from the American Type Culture Collection. The strains of fungi were inoculated on a medium containing 30 mL of potato dextrose agar and placed in an incubator at 28 °C for 72 hours to obtain mature microbial spores (mycelium). This process was carried in a BIOTECH-30SS solid fermentation tank (Shanghai Baoxing Biological Engineering Equipment Co., Ltd). A KQ-C type automatic steam generator (Shanghai Fengxian Xiexinji Power Plant) was used to generate steam for sterilization, and 2 kg of air-dried corn straw (particle size = 0.5 cm) was sterilized in a solid fermenter. The sterilization process was conditioned for 25 min at 121 °C. After sterilization, the *T. reesei* liquid containing the spore mycelia (0.8 L) and a mineral salt solution (5 L) was mixed with sterilized corn straw. The spore solution and a mineral salt solution (pH = 5) used were prepared similarly as described by Zhang et al (2020), and the C/N ratio was adjusted to 25:1 using a mineral salt nutrient solution. The mineral salt nutrient solution (g $L^{-1}$) was prepared as a mixture of: $KH_2PO_4$ 28 g, $(NH_4)_2SO_4$ 9.6 g, $MgSO_4$ 4.2 g, $CoCl_2$ 4.2 g, $(NH_2)_2CO$ 2.2 g, $FeSO_4 \cdot 7H_2O$ 0.07 g, $CaCl_2$ 0.028 g, $MnSO_4$ 0.021 g, $ZnSO_4$ 0.019 g, and the pH = 5. The fermentation process lasted 90 days and was carried out at 30 °C, 60% humidity, and 6.0 rpm. The final fermented product after 90 days was designated as fermented corn straw treated with *T. reesei* (FCS-T) material. The basic elemental properties of the CS and FCS-T materials are presented in **Table 2**, and were determined with an element analyzer (Vario-EL-III Hanau, Germany).

**2.3 Field procedures and sampling**

**2.3.1 Experimental layout: Field plot settings and specifications**

The field experiment was set up to have nine plots and three treatments, namely CS, FCS-T, and CK (as a control), which were applied under equal C mass. Each treatment was replicated three times and arranged in a completely randomized design. The size of each plot was 0.6 m × 0.6 m. The specific scheme of soil treatment is shown in Figure 1.

The CS return treatment was prepared by mixing 360 g of corn straw residues (equivalent to 1 kg $m^{-2}$) in the 0 – 20 cm surface soil layer, and exactly 5.975 g $CH_4N_2O$ was added to adjust the C/N ratio to 25:1 (which is suitable for soil microbial growth (Chapin III et al., 2011)). Thereafter, base fertilizer (17.68 g of $CH_4N_2O$ and 7.92 g of $KH_2PO_4$) was applied to the 0 – 20 cm soil layer.

Preparation of FCS-T treatment was done by mixing 428 g (the same amount of C mass as the C of the CS material) of fermented corn straw treated with *T. reesei* material (equivalent to 1.189 kg $m^{-2}$) in the topsoil layer of 0–20 cm. The same amount of base fertilizer as in the CS treatment (17.68 g of $CH_4N_2O$ and 7.92 g of $KH_2PO_4$) was applied in the 0–20 cm depth of the FCS-T plots.

The blank control (CK) treatment was prepared by only mixing 17.68 g of $CH_4N_2O$ and 7.92 g of $KH_2PO_4$ fertilizer to the 0–20 cm soil depth.

### 2.3.2 Soil sampling and analysis

Five topsoil samples (0 – 20 cm) were collected from each plot at 0 d, 30 d, 60 d, 90 d, 180 d, and 360 d using a stainless-steel soil auger (5 cm in diameter). For each soil sampling day, all visible corn straw materials in CS and FCS-T soils were picked out with tweezers and returned to their respective plots. The collected fresh soil was immediately divided into two sub-samples and passed through a 2 mm sieve. One subsample was then placed in a refrigerator (4 $\circ$C) to later analyze MBC in soil. The remaining subsample was air-dried to determine SOC, EOC, WEOC content and humus composition.

### 2.4 Analytical methods

#### 2.4.1 Labile soil organic carbon fractions

The SOC content was determined by the potassium dichromate oxidation method (Nelson and Sommers, 1982). The WEOC content was obtained by successively extracting 5 g of air-dried soil samples with distilled water in a 1:6 ratio of soil to water. The soil-solution mixture was shaken on a reciprocal shaker at 25 $\circ$C for 60 min, and then centrifuged at 4500 rpm for

20 min. The solution was filtered through a 0.45-μm filter membrane (Changtingny et al., 2010). The EOC content was determined using the $KMnO_4$ (333 mM) oxidation procedure (Lefroy et al., 1993). Fresh soil equivalent to 10 g of oven-dried soil was fumigated with $CHCl_3$ for 24 h and the other 10 g of soil was not fumigated. Both fumigated and unfumigated soils were then extracted with 0.5 mol $L^{-1}$ $K_2SO_4$. The MBC content was estimated from the increase in organic C in the 0.5 mol $L^{-1}$ $K_2SO_4$ extracts of $CHCl_3$ fumigated soils as described by Vance et al. (1987). The soil WEOC and MBC contents

were determined by a TOC analyser (Shimadzu TOC-VCPH, Japan). MBC was calculated as below Eq. (1):

$$MBC = \frac{F_c}{k_c} \qquad (1)$$

where $F_c$ is the difference between the amount of $CO_2$ released by fumigated and unfumigated soil (control) during the cultivation period; $k_c$ is the conversion coefficient.

The carbon available ratio (CAR) of labile organic C contents (WEOC, EOC and MBC) were calculated as below Eqs.

(2.3.4):

$$CAR_{(WEOC)} = \frac{WEOC\ (mg\ kg^{-1})/1000}{SOC\ (g\ kg^{-1})} \times 100\% \qquad (2)$$

$$CAR_{(EOC)} = \frac{EOC\ (g\ kg^{-1})}{SOC\ (g\ kg^{-1})} \times 100\% \qquad (3)$$

$$CAR_{(MBC)} = \frac{MBC\ (mg\ kg^{-1})/1000}{SOC\ (mg\ kg^{-1})} \times 100\% \qquad (4)$$

The carbon pool index (CPI), carbon pool activity (CPA), carbon pool activity index (CPAI) and carbon management pool

index (CPMI) were calculated, according to Blair et al. (1995) and Jiang et al. (2021) as below Eq. (5):

$$CPI = \frac{SOC_{Treatment}}{SOC_{CK_0}} \tag{5}$$

where $SOC_{Treatment}$ represents the SOC content (g kg$^{-1}$) in soil of a given treatment (CS, FCS-T or CK), $SOC_{CK_0}$ represents the SOC content (g kg$^{-1}$) in soil of CK at 0 d.

$$NLOC = SOC - EOC \tag{6}$$

NLOC represents the non-labile organic C content (g kg$^{-1}$), which is the difference between the SOC content and EOC content.

$$CPA = \frac{EOC}{NLOC} \tag{7}$$

$$CPAI = \frac{CPA_{Treatment}}{CPA_{CK_0}} \tag{8}$$

where $CPA_{Treatment}$ represents the CPA in soil of a given treatment (CS, FCS-T or CK), $CPA_{CK_0}$ represents the CPA in soil of CK at 0 d.

$$CPMI = CPI \times CPAI \times 100 \tag{9}$$

### 2.4.2 Humus composition

Humus composition was sequentially analyzed following the International Humic Substances Society procedure (Kumada 1987) described in detail by Dou (2010). Briefly, 5 g of air-dried soil was extracted with a 30 mL mixture of 0.1 M alkali solution (NaOH + Na$_4$P$_2$O$_7$) under permanent shaking at 70 ∘C for 1 h and centrifuged. The remaining soil residue was humin, and the mixture, which is humus extract, was acidified with 0.5 M sulfuric acid to separate humic acid and fulvic acid. The carbon contents of the humus extract (HE-C), humic acid (HA-C) and humin (HM-C) were determined. Then the C content of fulvic acid (FA-C) was calculated as the difference between HE-C and HA-C. The humification degree (PQ) was calculated as HA-C/HE-C ratio (Sugahara and Inoko, 1981).

### 2.5 Statistical analysis

Microsoft Office Excel 2017 was used for data processing, and the statistical analysis was performed by SPSS Statistics 22.0 (IBM Statistics 21.0). Significant differences among treatment means were evaluated using the least significant difference test with TUKEYs adjustment at $P < 0.05$. Principal component analysis (PCA) was performed with Minitab 18 software (Pennsylvania, USA) to check for similarities between treatments. The graphs were compiled using the Origin 2019 software (OriginLab Corporation).

## 3 Results

### 3.1 Changes in SOC contents

At 0 d, the SOC content did not differ significantly between the three treatments, but differed significantly from 30 d to 360 d among the three treatments (**Figure 2**). Comparing all treatments, the FCS-T treatment showed significantly higher SOC content, whereas the CK had significantly lower SOC content throughout the study period. The CS and FCS-T treatments showed the largest increase in SOC content with the increase in the duration of the study. Whereas, SOC content in the CK treatment did not change significantly throughout the 360-day period. At the 360 d, the SOC content of CS and FCS-T was 12.71% and 18.81% higher compared with that of CK, respectively.

### 3.2 Changes in soil labile organic carbon fractions and carbon available ratios

In the 360-day field experiment, the WEOC, EOC and MBC contents of CK, CS and FCS-T treatments showed a similar changing trend (**Figure 3**). The content of these attributes firstly increased from 0 d to the 90 d, and then gradually decreased to the 360 d in the CS and FCS-T treatments. Water extractable organic C, EOC and MBC contents of CS and FCS-T treatments were highest at 90 d. The WEOC, EOC and MBC contents of CK appeared to slightly decrease with the duration of the experiment. Comparing all treatments, the contents of WEOC, EOC and MBC did not differ significantly at 0 d, 30 d, 180 d, and 360 d between CS and FCS-T treatments.

In terms of WEOC, the carbon available ratios of CS (1.19 %) and FCS-T (1.29 %) treatments was highest at 60 d, and lowest at 0 d (**Table 3**). In terms of EOC, the carbon available ratios of CS (9.25 %) and FCS-T (9.34 %) treatments was significantly higher at 90 d. With respect to MBC, the carbon available ratio of FCS-T (5.80 %) treatment was also higher at 90 d and that of CS (2.92 %) treatment was significantly higher at 60 d. Irrespective of sampling time, the carbon available ratios of WEOC, EOC and MBC was always significantly higher under FCS-T and CS treatments compared with CK. These parameters did not always differ significantly between the CS and FCS-T treatments.

### 3.3 Soil carbon pool management index

The soil carbon pool management index was computed at the end of the experiment (i.e., day 360). At the 360th day, the CS and FCS-T treatments significantly increased thecarbon pool management index and carbon pool activity index compared with the CK treatment, but the carbon pool activity index of CS and FCS-T treatments did not differ significantly (**Table 4**). Applying CS and FCS-T significantly increased the carbon pool management index compared with CK, increasing the CPMI by 17.3 % and 31.7 %, respectively.

### 3.4 Humus composition and C content in soil under different treatments

At 0 d of the experiment, there was no significant difference in the relative content (**Table 5**) and composition of humus C among the three treatments (**Figure 4**). With the application of CS and FCS-T, the HE-C and HM-C contents in the soil

increased with the duration of the experiment. Compared with CK, the CS and FCS-T treatments increased the HE-C content in soil. At 360 d, the HE-C content of the FCS-T treatment was significantly higher than that of the CS treatment, and the HE-C of the FCS-T and CS treatments increased by 1.99 g kg$^{-1}$ and 1.31 g kg$^{-1}$, respectively, when compared with that at 0 d. The HM-C content of the FCS-T treatment increased significantly when compared with other treatments over the duration of the experiment, with a cumulative increase of 0.79 g kg$^{-1}$ at 360 d (**Figure 4**). Throughout the duration of the experiment, there was no significant difference observed between CS and CK treatments with respect to HM-C content.

Compared with CK, application of CS and FCS-T increased the FA-C content in the soil with the duration of the experiment (**Figure 5**). The highest FA-C content in the CS and FCS-T treatments was measured at 180 d, and the lowest FA-C content was recorded at 0 d. The content of HA-C under CS and FCS-T treatments increased with the duration of the experiment. The highest HA-C content in the CS and FCS-T treatments was measured at 360 d, and the lowest HA-C content was recorded at 0 d. The content of HA-C in the FCS-T treatment at 360 d was 0.77 g kg$^{-1}$ higher than that in the CS treatment.

### 3.5 Multivariate Analysis

The relationship between SOC parameters and humus components, shown according to PCA (**Figure 6**), was well confirmed by Pearson's correlation analysis (**Figure 7**). Figure 6 indicated that under all the treatments, the HA-C, HM-C, and EOC contents exhibited significant correlations with SOC content, carbon pool activity index, and carbon pool management index, whereas WEOC and MBC contents were significantly correlated with the FA-C content. The PCA clearly separated the three treatments, which implies that each treatment had a distinct influence on SOC content, CPMI, and humus component characteristics. The correlation between SOC and carbon pool management index was more pronounced under the CS and FCS-T treatments. The correlation between MBC and SOC was stronger under the CS treatment, and the correlations between WEOC, MBC and FA-C were more pronounced under the FCS-T treatment.

### 4 Discussion

### 4.1 Effects of different treatments on SOC, soil labile organic carbon fractions, and humus fractions

A large number of studies has shown that application of organic materials is beneficial to the accrual of SOC (Ros et al., 2006; Zhang et al., 2015b) and distribution of labile organic C components (Blair, 2000; Chen et al., 2009; Sainepo et al., 2018). This is consistent with the results of our study which showed that application of CS and FCS-T increased SOC content (**Figure 2**), MBC, WEOC and EOC contents (**Figure 3**). Although applied under equal C mass input, the FCS-T treatment appeared to sequester more organic C in the soil than the CS treatment. This may be because, the FCS-T used in the present study was produced by fermentation with *T. reesei* and had lower C/N ratio (**Table 2**). During the fermentation process, studies show that part of the organic matter input is converted into $CO_2$ and other substances, and the remaining residue is converted into stable organic matter similar to HS (Atiyeh et al., 2002; Romero et al., 2007). The closer the substrate's C/N ratio is to the microorganisms' C/N ratio, the more significant the fraction of substrate C that remains in the

soil (Hessen et al., 2004). Furthermore, according to Sprunger et al. (2019) low C/N ratio of organic residues promote the accumulation of soil organic matter. Whereas, organic inputs applied to the soil with a large C/N ratio such as the CS treatment in the case of our study, may lose more C in turnover compared with organic amendments with a small C/N ratio (Dannehl et al., 2017). The C/N ratio of organic amendments and the C fate in soil had a negative connection (Dannehl et al., 2017) The aforesaid point of view was further supported by our research.

Our results further showed that after FCS-T and CS application, the concentrations of WEOC, EOC, and MBC in the soil increased at the initial stages of the experiments (i.e., 0 – 90 d), and then gradually decreased towards the end of the experiment (**Figure 3**). In contrast, the HE-C and HM-C contents appeared to increase with the duration of the experiment, with the greater increase reported in the FCS-T treatment (**Figure 4** and **5**). This result is consistent with the findings of Guan et al. (2015). The reason for this phenomenon may be that the WEOC and EOC are easily and the first organic compounds to be utilized by soil microorganisms (Haynes et al., 2005). Corn straw contains aromatic C compounds (Roldán et al., 2011; Zhang et al., 2020), which are more difficult to decompose and tend to accumulate as humic substances (Kuzyakov et al., 2009; Pan et al., 2016; Dou et al., 2020).

Comparing all treatments, the FCS-T treatment appeared to have significantly higher WEOC content than CS, but the EOC content did not always differ significantly between the FCS-T and CS treatments during the duration of the experiment (**Figure 3**). Ma et al. (2021) reported similar findings with barley treated with microbial inoculant, in which the WEOC content was significantly higher than that of barley residue without microbial inoculant, but the EOC content differed seldomly. The higher EOC content in FCS-T treatment than that in CS treatment (**Table 3**), suggests that organic matter after microbial treatment is likely converted into EOC. During the entire duration of the present experiment, the MBC content of FCS-T treatment was also higher than that of CS treatment, but not always significant. Ng et al. (2016) reported similar observations, and this may be due to the fact that crop residues treated with microbial inoculants are easily assimilated by soil microorganisms (Gaind and Nain, 2006; Vargas-Garcia et al., 2006; Pan et al., 2016). Thereby, promoting the sequestration of organic C in organic materials.

The WEOC and EOC of the soil largely depend on the SOC content (Guan et al., 2018). This was also confirmed by the results from the present study, which showed SOC content to be positively correlated with WEOC, EOC and MBC contents (**Figure 7**). This means the WEOC, EOC, and MBC can be used as the best proxies to detect changes in SOC content, since these fractions respond promptly to changes in soil management practices. In the present study, correlation of SOC content with MBC and EOC was more pronounced under FCS-T treatment than CS and CK treatments. This may be likely due to differences in the chemical composition of these treatments. Vanlauwe et al. (2005) and Mandal et al. (2007) found that changes in soil C is mainly influenced by the chemical composition of the applied organic matter.

Compared with other treatments, the FCS-T treatment significantly increased the contents of HA-C and FA-C, throughout the 360 d period (**Figure 5**). This is because the FCS-T material contain relatively higher alkyl, aromatic C contents, and humic-like substances (see Zhang et al., 2021); compounds that are resistant to microbial direct degradation. This promoted the formation of soil humic substances during the process of humification (Huang et al., 2008; Roldán et al., 2011). The parts

of the humic-like substances in the FCS-T treatment gradually formed new FA and HA in the soil possibly through microbial *ex vivo* modification and *in vivo* turnover (Liang et al., 2017). Zhang et al. (2019) further found that application of fermented corn straw was more conducive to the increase of HA-C, and CS was more conducive to the increase of FA-C. In other studies, Gaind and Mathur (2001) and Gaind and Nain (2007) reported statistically significant increasing humus content in soil treated with paddy/wheat straw compost with *T. reesei* under rice-wheat cropping system. In our study, by analyzing the changes of humic substances components in different time periods, we determined that administration of FCS-T can significantly and continuously increase FA relative content up to 180 days, while the increase in HA relative content can last up to 360 days (**Table 5**). The FA increased in the early stage may be converted into HA and stored in the soil in the later stage. Therefore, application of FCS-T materials is more conducive to increasing humus C content and HA relative content (include more aromatic C compounds), which is important for long-term storage of SOC.

### 4.2. Relationships between SOC, soil labile organic carbon fractions, humus components and CPMI

The results of this study showed that the increase in SOC content was mainly due to the increase in EOC, HA-C, and HM-C contents, rather than the accumulation of WEOC and MBC. The possible explanation is that WEOC and MBC are more easily utilized by soil microorganisms, and their ratio in SOC is much lower (Blair, 2000; Haynes, 2005). The carbon pool management index is a comprehensive index to evaluate SOC variation rates in response to soil management practices. For instance, a high carbon pool management index indicates that the soil management practices have a stronger potential to promote soil C sequestration (Blair et al., 1995). In the present study, the FCS-T treatment showed significantly higher carbon pool management index and carbon pool index than CS and CK treatments (**Table 4**). This suggests that the FCS-T treatment was more conducive to the accumulation of organic C in the soil. This result may be due to the fact that soil C accumulation is mainly driven by increased plant residue input, which increases SOC content. The MBC was positively correlated with FA-C and SOC (**Figure 6** and **7**), providing evidence that the activity of microorganisms affects the accumulation of FA in the soil, thereby promoting the increase of SOC.

### 5 Conclusion

In this 360-day field experiment, we applied corn straw (CS) and fermented corn straw treated with *Trichoderma reesei* (FCS-T) under equal C input, and a blank control treatment (CK) for comparison. The following conclusions were drawn:

1. The FCS-T treatment was more effective than CS treatment in terms of soil carbon storage. The SOC content of CS and FCS-T treatments increased by 12.71% and 18.81%, respectively, when compared with CK at the 360-d. Contrasted with direct application of CS, the FCS-T treatment increased SOC reserve by 1.715 g kg$^{-1}$.

2. In terms of soil humus content and humification, FCS-T treatment was more effective than CS and CK treatment in accelerating the accumulation of HA-C and FA-C content in the soil. The relative content of HA-C in the FCS-T treatment was 1.9% higher than that in CS treatment at 360 days, and the PQ value of FCS-T increased to 74.1%. The FCS-T treatment

was more conducive to the accumulation of stable HM-C component in the soil, with a cumulative increase of 0.79 g kg$^{-1}$ at day 360.

3. The application of FCS-T in the soil resulted in an increase in the labile organic carbon fractions and carbon pool management index, which was more pronounced on the 60$^{th}$ and 90$^{th}$ days. Amongst all labile soil organic carbon fractions, the FCS-T treatment is more conducive with the increase of WEOC content. The CS and FCS-T treatments had similar effects on the carbon available ratio of the soil labile organic carbon fractions. The FCS-T treatment was more advantageous with increasing the content of HA-C, HM-C and WEOC, which resulted to the overall increase in SOC and EOC, as well as the carbon pool management index.

The results confirmed our initial hypothesis that the application of FCS-T has a greater potential to increase soil carbon sequestration compared with direct application of CS. As a method of returning straw residues to the field, the application of FCS-T is a practice worthy of further exploration.

**Data availability**

The data generated in this study are available from the corresponding authors upon reasonable request.

**Author Contribution**

Yifeng Zhang: Conceptualization, Methodology, Software, Data curation, Writing-Original draft preparation; Sen Dou: Formal analysis, Funding acquisition, Supervision; Batande Sinovuyo Ndzelu: Validation, Writing-review & editing; Rui Ma: Supervision; Dandan Zhang: Data curation; Xiaowei Zhang: Supervision; Shufen Ye: Data curation; Hongrui Wang: Data curation.

**Competing interests**

The contact author has declared that neither they nor their co-authors have any competing interests.

**Funding**

The study was supported by the National Natural Science Foundation of China (42077022) and the Key Research and Development Program of Jilin Province (20200402098NC)

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

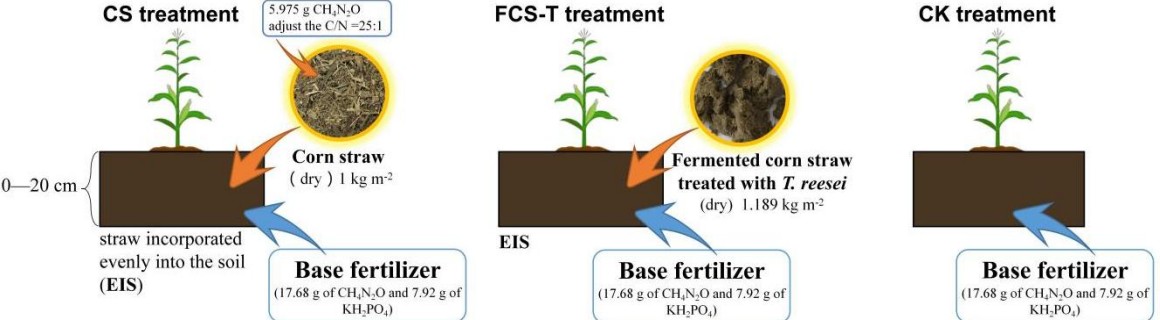

**Figure 1: Schematic diagram of three different treatment methods in the field**

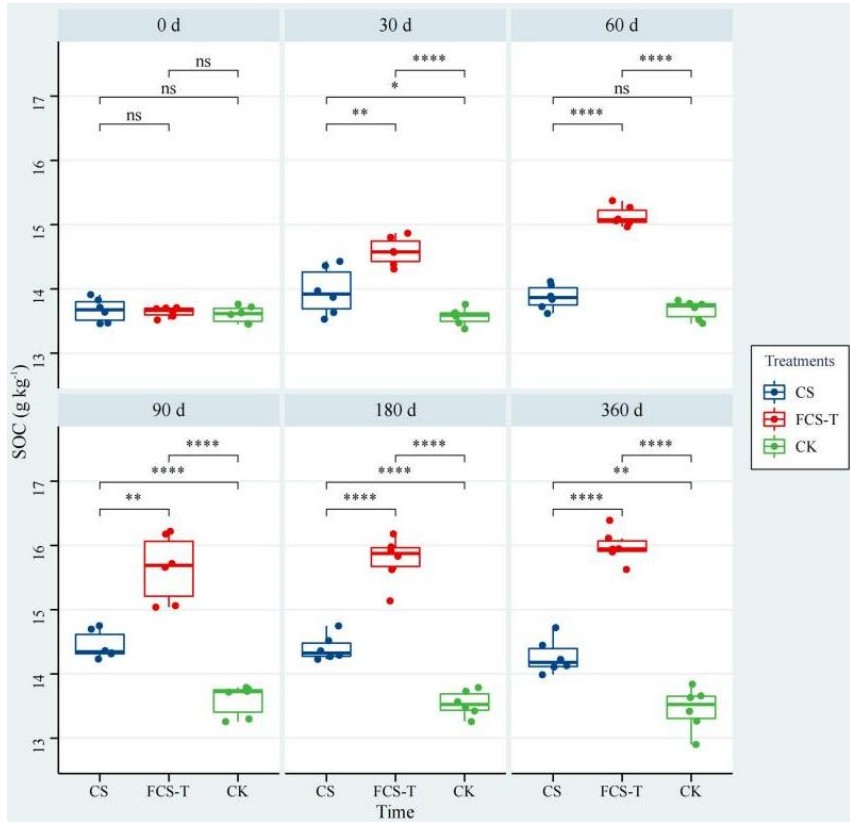

**Figure 2: Soil organic carbon content of the three different treatments during the 360-d experimental period.**

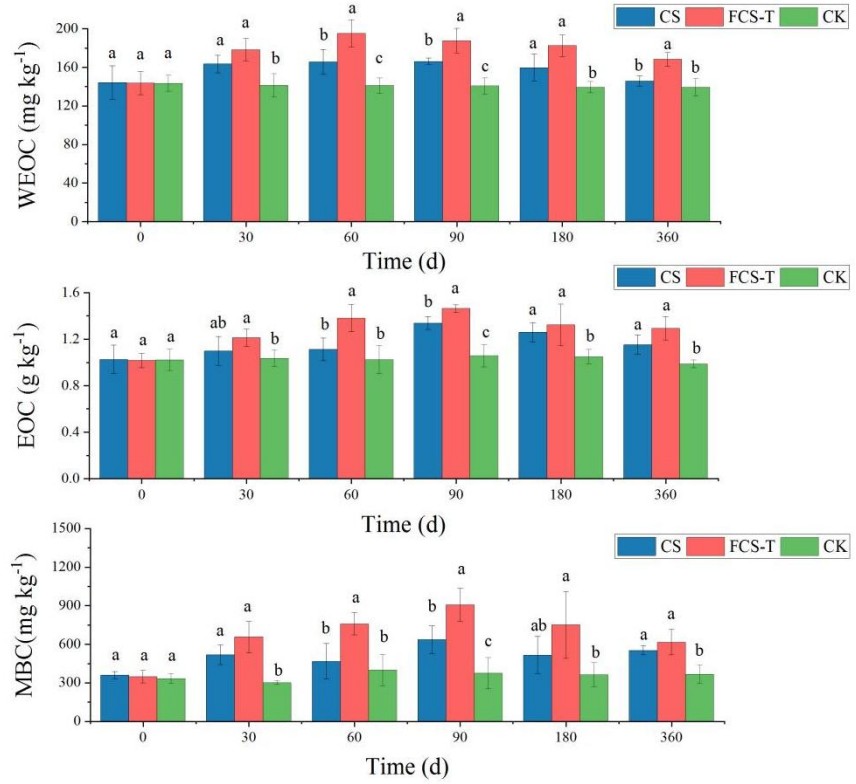

**Figure 3: Effects of corn straw returned (CS), fermented corn straw treated with *T. reesei* returned to the field (FCS-T) and non-straw amended soil (CK) on soil labile organic carbon (WEOC, EOC and MBC) concentrations in the 0–20 cm soil depth.**

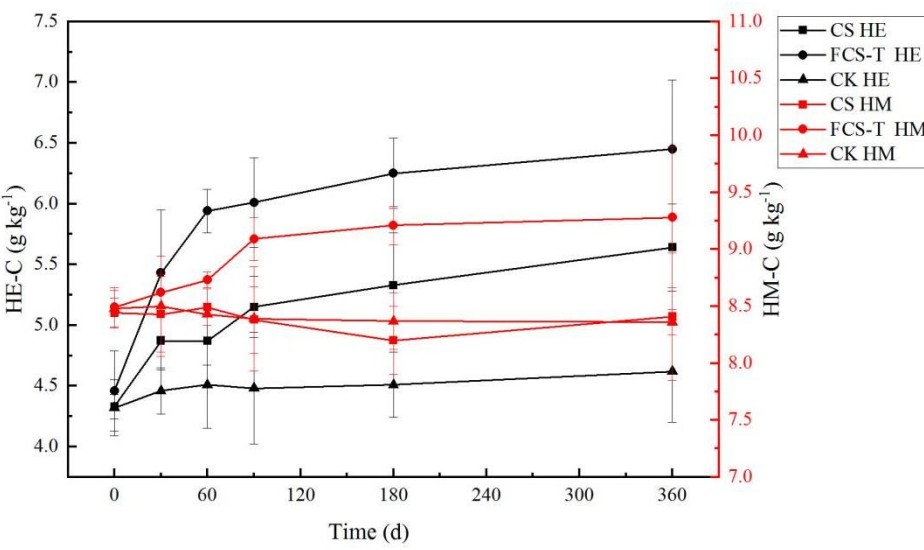

**Figure 4: The carbon content of humus extracted (HE-C) and humin (HM-C) of the three different treatments during the 360-day**

**period.**

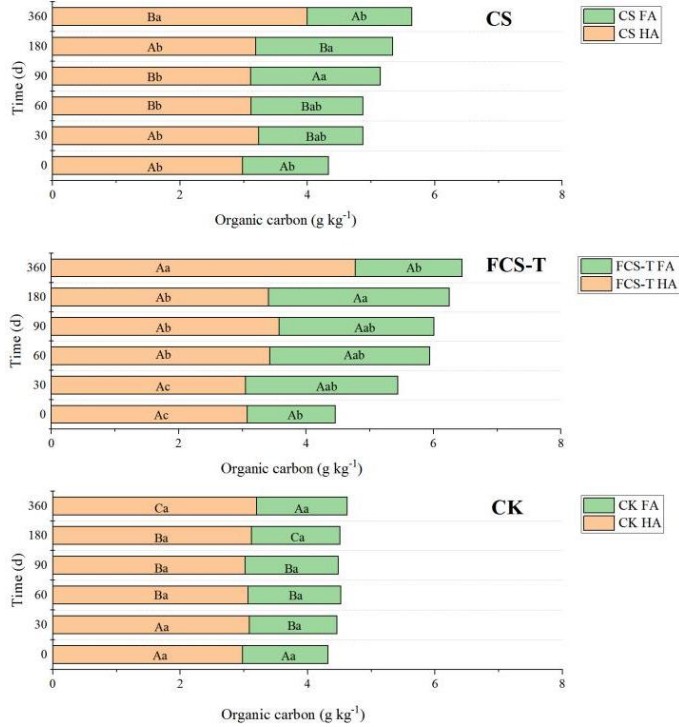

**Figure 5: The carbon contents of humic acid (HA) and fulvic (FA) isolated from different treatments.**

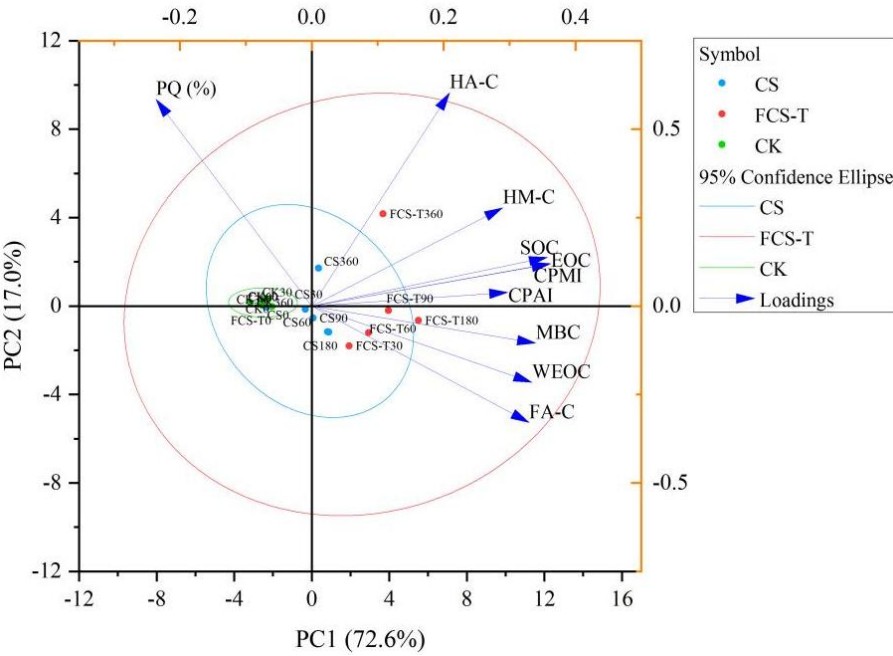

**Figure 6: Principal component analysis (PCA) of soil organic carbon parameters and humus components affected by different treatments during 360 d of the experiment.**

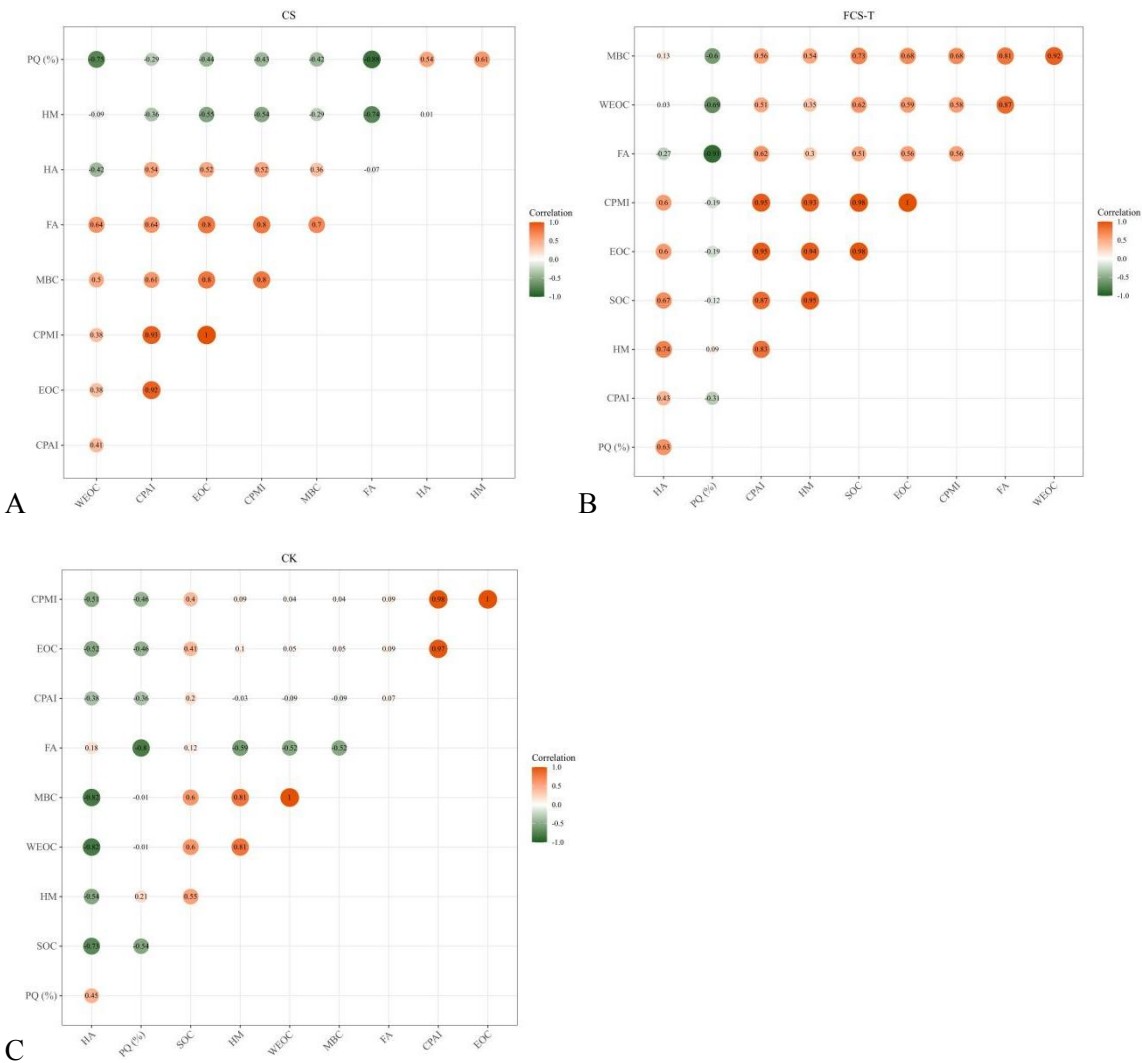

**Figure 7:** The heatmap of Pearson's correlations among soil parameters from different treatments soil during the 360 d of the experiment. (A, CS; B, FCS-T; C, CK).

**Figure captions**

**Figure 1.** Schematic diagram of three different treatment methods in the field.

**Figure 2.** Soil organic carbon content of the three different treatments during the 360-d experimental period. The upper, middle, and lower horizontal lines of the box represent the upper quartile, the median, and the lower quartile, respectively. The values represented by the upper- and lower-line segments refer to the maximum and minimum values of the data, and

525 the points outside the box represent outliers. The symbol on the figure indicates the $P$ value between two variables. The number of "*" indicates the degree of significance. For example: "*" means $P < 0.05$, "**" means $P < 0.01$, "***" means $P <$

0.001, "****" means $P < 0.0001$, and "ns" means no significance. CS, corn straw returned to the field; FCS-T, fermented corn straw treated with *T. reesei* returned to the field; CK, blank control treatment.

**Figure 3.** Effects of corn straw returned (CS), fermented corn straw treated with *T. reesei* returned (FCS-T) and non-straw amended soil (CK) on soil labile organic carbon (WEOC, EOC and MBC) concentrations in the 0−20 cm soil depth. Each bar represents the mean ± standard deviation in the figure (n=3). Different lowercase letters within the same time indicate significant differences among different treatments at P < 0.05 level.

**Figure 4.** The carbon content of humus extracted (HE-C) and humin (HM-C) of the three different treatments during the 360-day period. Error bar is the standard deviations of triplicate averages. CS, corn straw returned to the field; FCS-T, fermented corn straw treated with *T. reesei* returned to the field; CK, blank control treatment.

**Figure 5.** The carbon contents of humic acid (HA) and fulvic (FA) isolated from different treatments. Each bar represents the mean of HA and FA in the figure (n=3). Different uppercase letters mean significant difference (P < 0.05) for either HA or FA among the treatments in a particular time (d) of the experiment. Different lowercase letters mean significant difference (P < 0.05) for either HA or FA among different experimental time (d) for a given treatment.

**Figure 6.** Principal component analysis (PCA) of soil organic carbon parameters and humus components affected by different treatments during 360 d of the experiment. (A: PCA bi-plot in 0-360 d; B: PCA bi-plot in the 360 d). Notes: PC, principal component; SOC, soil organic carbon; CPMI, carbon pool management index; CPAI, carbon pool activity index; WEOC, dissolved organic carbon; EOC, easily oxidizable organic carbon; MBC, microbial biomass carbon; HA, humic acid; FA, fulvic acid; HM, humin; PQ, humification degree. CS, corn straw returned to the field; FCS-T, fermented corn straw treated with *T. reesei* returned to the field; CK, blank control treatment.

**Figure 7.** The heatmap of Pearson's correlations among soil parameters from different treatments soil during the 360 d of the experiment. (A, CS; B, FCS-T; C, CK).

**Table 1.** Basic properties of the soil in field experiments

| Soil | pH | Organic matter (g kg$^{-1}$) | Alkaline N (mg kg$^{-1}$) | Available P (mg kg$^{-1}$) | Available K (mg kg$^{-1}$) |
|---|---|---|---|---|---|
| Black soil | 6.55±0.31 | 51.18±1.41 | 7.44±0.57 | 565.0±2.3 | 59.00±0.85 |

Note: Values (± standard deviation) were averaged over 3 replicates.

**Table 2.** Elemental composition of materials used in field experiments

| Materials | C (g kg$^{-1}$) | H (g kg$^{-1}$) | N (g kg$^{-1}$) | O (g kg$^{-1}$) | C/N |
|---|---|---|---|---|---|
| CS | 376.4±1.0 | 51.18±0.33 | 7.44±0.03 | 565.0±0.8 | 50.57±0.08 |

| FCS-T | 319.4±1.4 | 43.87±0.25 | 29.50±0.12 | 607.2±1.5 | 10.83±0.09 |

Note: Values (± standard deviation) were averaged over 3 replicates. CS, corn straw; FCS-T, fermented corn straw treated with *T. reesei*.

555

**Table 3.** The carbon available ratio (CAR) of dissolved (WEOC), easily oxidizable organic carbon (EOC), and microbial biomass carbon (MBC) under different treatments in the 0-360 d period.

| CAR (%) | WEOC | | | EOC | | | MBC | | |
|---|---|---|---|---|---|---|---|---|---|
| Time (d) | CS | FCS-T | CK | CS | FCS-T | CK | CS | FCS-T | CK |
| 0 | 1.06a | 1.05a | 1.06a | 7.50a | 7.45a | 7.51a | 2.63a | 2.55a | 2.45a |
| 30 | 1.17a | 1.22a | 1.04b | 7.88b | 8.31a | 7.62b | 3.73b | 4.51a | 2.22c |
| 60 | 1.19b | 1.29a | 1.03c | 8.02b | 9.11a | 7.48c | 3.38b | 5.01a | 2.92c |
| 90 | 1.15b | 1.20a | 1.04c | 9.25a | 9.34a | 7.78b | 4.40b | 5.80a | 2.76c |
| 180 | 1.11a | 1.15a | 1.03b | 8.76a | 8.31b | 7.75c | 3.60b | 4.72a | 2.68c |
| 360 | 1.02b | 1.05a | 1.04a | 8.06a | 8.08a | 7.34b | 3.88a | 3.86a | 2.73b |

Note: Values are means that do not share the same letter for a given parameter and time (d) of experiment are significantly different ($P < 0.05$). CS, corn straw returned to the field; FCS-T, fermented corn straw treated with *T. reesei* returned to the field; CK, blank control treatment.

560

**Table 4.** The carbon management indices under different treatments during the 360-d experimental period.

| Indexes | Treatments | values |
|---|---|---|
| CPI | CS | 1.049±0.029b |
| | FCS-T | 1.175±0.028a |
| | CK | 0.989±0.036c |
| CPA | CS | 0.088±0.006a |
| | FCS-T | 0.088±0.005a |
| | CK | 0.079±0.004a |
| CPAI | CS | 1.080±0.068a |
| | FCS-T | 1.083±0.065a |
| | CK | 0.977±0.054b |
| CPMI | CS | 113.20±4.48b |
| | FCS-T | 127.15±5.57a |
| | CK | 96.51±3.47c |

**Table 5.** Changes of relative content of each humic substances component and the PQ values in different treatments during the 0-360 d period.

| Time (d) | Treatments | HA (%) | FA (%) | HM (%) | PQ (%) |
|---|---|---|---|---|---|
| 0 | CS | 21.8±1.5a | 9.9±1.1a | 61.8±1.3a | 68.9±3.7a |
| | FCS-T | 22.6±0.9a | 10.3±2.1a | 62.3±1.3a | 69.2±4.5a |
| | CK | 21.9±0.3a | 9.9±2.2a | 62.3±1.5a | 69.1±4.9a |
| 30 | CS | 23.3±1.2a | 11.7±1.9b | 60.6±1.4a | 66.7±4.7a |
| | FCS-T | 20.9±0.8b | 16.3±2.4a | 59.1±1.8a | 56.2±3.1b |
| | CK | 22.8±0.9a | 10.1±1.5b | 62.6±4.1a | 69.3±4.7a |
| 60 | CS | 22.5±0.6a | 12.6±1.6b | 61.2±0.1a | 64.2±3.6a |
| | FCS-T | 22.7±0.6a | 16.6±1.8a | 57.7±1.0b | 57.8±3.3b |
| | CK | 22.4±0.8a | 10.6±1.7b | 61.7±1.2a | 68.1±2.8a |
| 90 | CS | 21.5±1.4a | 14.1±2.3ab | 58.0±1.6a | 60.55±4.8b |
| | FCS-T | 22.8±1.6a | 15.5±1.8a | 58.1±1.0a | 59.54±4.4b |
| | CK | 22.3±1.8a | 10.8±1.4b | 61.8±4.4a | 67.42±1.0a |
| 180 | CS | 22.2±0.7a | 14.9±1.6b | 57.1±2.5b | 59.8± 2.4b |
| | FCS-T | 21.4±1.6a | 17.9±1.8a | 57.9±1.4b | 54.6± 3.1b |
| | CK | 23.0±0.9a | 10.3±0.5c | 61.9±1.9a | 69.2± 0.8a |
| 360 | CS | 28.0±1.9a | 11.5±1.3a | 58.9±2.6b | 71.0±1.4ab |
| | FCS-T | 29.9±2.2a | 10.5±1.7a | 58.0±3.3b | 74.1±1.8a |
| | CK | 23.8±1.2b | 10.6±1.1a | 62.2±1.9a | 69.2±1.7b |

Note: Values are means ± SE. Means that do not share the same letter within a column of a given parameter and time (d) of the experiment are significantly different ($P < 0.05$). HA, humic acid; FA, fulvic acid; HM, humin; CS, corn straw returned to the field; FCS-T, fermented corn straw treated with *T. reesei* returned to the field; CK, blank control treatment.