# Peer review of "Effects of returning corn straw and fermented corn straw to fields on the soil organic carbon pools and humus composition"

_SOIL, 2021_

## Author Response (AR1)

**Soil**
**Manuscript No.:** SOIL-2021-105
**Manuscript Title:** Effects of returning corn straw and fermented corn straw to fields on the soil organic carbon pools and humus composition
**Article Type:** Research paper
**Authors:** Yifeng Zhang, Sen Dou, Batande Sinovuyo Ndzelu, Rui Ma, Dandan Zhang, Xiaowei Zhang, Shufen Ye, Hongrui Wang

**Topical Editor Comments to the author**

Dear Dr. Yifeng Zhang,

Thank you for submitting your manuscript to SOIL.
I have revised your manuscript following the interactive discussion with referees. The manuscript is valuable and of interest for SOIL readers due to the important focus on organic matter stabilization as affected by corn straw management. However, the work does not add specific novelty to the current state of knowledge and Introduction seems to be a textbook rather than a focus on the main state of the art while highlighting the gaps that lead to the main objectives of the work. The two reviewers reported also some criticisms in the description of results and in the Discussion, that were well replied by the Authors in the Open discussion.
Based on these considerations the Authors are asked to revise the manuscript accordingly and submit a new, strongly revised version of the manuscript.

Best regards,
Luisella Celi

**Response to Topical Editor comments**

Dear Editor Luisella Celi,

First of all, we would like to thank you for your and reviewers' valuable comments and suggestions which help us to improve our manuscript. The manuscript has certainly benefited from these insightful revision suggestions.
Thank you very much for your encouragement and affirmation of the value and interest of our manuscript. We have deleted the textbook-style expressions, and significantly revised the Introduction section, reformulate a clearer research hypothesis, and put more effort into showcasing the latest technology in crop residue recycling as soil amendments. We also tried to show and explain the novelty of our research. Meanwile, we greatly revised the Discussion section. We hope that our new, strongly revised version of the manuscript successfully addressed each reviewer's questions point by point.

Yours sincerely,
Dr. Yifeng Zhang

**Response to the first reviewer's comments**

First of all, we would like to thank you for your valuable comments and suggestions which help us to improve our manuscript. Below we try to address all the points which you have indicated in your assessment opinions.

**RC1: 'Comment on soil-2021-105', Anonymous Referee #1, 24 Nov 2021**

**General comment**

➢ **Comment:** The topic of the manuscript titled "Effects of returning corn straw and fermented corn straw to fields on the soil organic carbon pools and humus composition" is of interest for the "SOIL" readership.

*Response:* Thank you very much for your support of our manuscript. We further revised our manuscript according to your comments. We have revised the manuscript carefully, and all changes in the revised manuscript are made using Track Changes to make reviewing easy.

**Specific comments**

(page, line: comment)

➢ **Comment 1:** 1, 24: Please write the acronym SOC here instead of on line 27

*Response:* This suggestion has been adopted. We have revised the acronym SOC as follows (**page 1, line 24-25**):

Text: "Recycling and returning crop residues as soil amendments is an important prospect for increasing soil organic carbon (SOC) content and crop yield (Villamil et al., 2015), as well as managing crop straw residues. "

➢ **Comment 2:** 3, 72: There is a new reference for the Soil Survey Staff. The USDA recommended citation is the following: Soil Survey Staff. 2014. Keys to Soil Taxonomy, 12th ed. USDA-Natural Resources Conservation Service, Washington, DC

*Response:* Thank you for the suggestion. We have made revisions and updates in the text and references as follows (**page 3, line 90-91**):

Text: "Soils in the study area are classified as Argiudolls according to the United States Department of Agricultural Soil Taxonomy (Soil Survey Staff. 2014)."

References: "Soil Survey Staff.: Keys to Soil Taxonomy: 12th edition. Natural Resources Conservation Service, USDA-Natural Resources Conservation Service, Washington, DC."

➢ **Comment 3:** 3, 85: Authors should detail the mineral salt solution they mixed to the corn straw. This could have influenced the characteristics of the fermented corn straw. For example, it showed higher N content than the unfermented corn straw (Table 2).

*Response:* We thank the reviewer for the suggestion. We have detailed the mineral

salt solution that was mixed with the corn straw as follows **(page 4, line 108-110**):

Text: "The mineral salt nutrient solution (g $L^{-1}$) was prepared as a mixture of: $KH_2PO_4$ 28 g, $(NH_4)_2SO_4$ 9.6 g, $MgSO_4$ 4.2 g, $CoCl_2$ 4.2 g, $(NH_2)_2CO$ 2.2 g, $FeSO_4·7H_2O$ 0.07 g, $CaCl_2$ 0.028 g, $MnSO_4$ 0.021 g, $ZnSO_4$ 0.019 g, and the pH = 5.

References: "Zhang, Y., Dou, S., Hamza, B., Ye, S., Zhang, D.: Mechanisms of three fungal types on humic-like substances formation during solid-state fermentation of corn straw, Intl. J. Agric. Biol., 24, 970 – 976, doi:10.17957/IJAB/15.1377, 2020b."

➢ **Comment 4:** 3, 93-105: Authors adjusted the C/N ratio of the corn straw residues to 25:1 adding urea. Apparently, they did not do the same procedure for the fermented corn straw, that showed a C/N ratio of about 10 (Table 2). Thus the mineralization of the two biomasses could have occurred differently also because of this parameter. Authors should consider also this when discussing their data.

*Response:* Thank you for the suggestion. In order to emphasize this point, we have modified the Discussion section. We added this content on the part of "4.1 Effects of different treatments on SOC and soil labile organic carbon fractions" to factor in the differences in the C/N ratio of the two materials used **(page 8, line 240 - page 9, line 246**):

Text: "The closer the substrate's C/N ratio is to the microorganisms' C/N ratio, the more significant the fraction of substrate C that remains in the soil (Hessen et al., 2004). Furthermore, according to Sprunger et al. (2019) low C/N ratio of organic residues promote the accumulation of soil organic matter. Whereas, organic inputs applied to the soil with a large C/N ratio such as the CS treatment in the case of our study, may lose more C in turnover compared with organic amendments with a small C/N ratio (Dannehl et al., 2017). The C/N ratio of organic amendments and the C fate in soil had a negative connection (Dannehl et al., 2017) The aforesaid point of view was further supported by our research."

References: "Hessen, D.O., Ågren, G.I., Anderson, T.R., Elser, J.J., de Ruiter, P .C.: Carbon sequestration in ecosystems: the role of stoichiometry, Ecology, 85, 1179 – 1192, doi: 10.2307/3450161, 2004.
Dannehl, T., Leithold, G., Brock, C.: The effect of C:N The relation between CUE and ratios on the fate of carbon from straw and green manure in soil. Eur. J. Soil. Sci., 68(6), 988 – 998, doi:10.1111/ejss.12497, 2017.
Sprunger, C. D., Culman, S. W., Palm, C. A., Thuita, M., Vanlauwe, B.: Long-term application of low C:N residues enhances maize yield and soil nutrient pools across Kenya, Nutr. Cycl. Agroecosyst., 114, 261 – 276, doi:10.1007/s10705-019-10005-4, 2019."

➢ **Comment 5:** 7, 210: Authors did not compost the corn straw residues, but they fermented it. Composting and fermentation are not exactly synonyms. Please correct here and throughout the paper.

*Response:* The suggestion of the reviewer was adopted. We changed the word "composting" to "fermentation" throughout the manuscript.

➢ **Comment 6:** Figure 1: It should show all details of treatments, i.e., the common base fertilization and the C/N ratio adjustment.

*Response:* Thank you for the suggestion. We have modified **Figure 1** and updated it in the manuscript. The revised version of **Figure 1** was as follows:

[Figure]

**Figure 1: Schematic diagram of three different treatment methods in the field**

➢ **Comment 7:** Table 1 and 2 should show g kg$^{-1}$ or mg kg$^{-1}$ instead of g/kg and mg/kg. Table 2 does not report any statistical analysis between the two biomasses.

*Response:* Thank you for the suggestion. We have corrected all similar errors in the pictures and tables. The statistical analysis label here in **Table 2** was forgotten by us. According to the raw data:

| | C (g kg$^{-1}$) | H (g kg$^{-1}$) | N (g kg$^{-1}$) | O (g kg$^{-1}$) | C/N |
|---|---|---|---|---|---|
| | 376.1 | 51.02 | 7.432 | 565.4 | 50.607 |
| CS | 377.5 | 50.96 | 7.478 | 564.1 | 50.482 |
| | 375.6 | 51.56 | 7.419 | 565.4 | 50.625 |
| stdev | 1.0 | 0.33 | 0.031 | 0.8 | 0.078 |
| average | 376.4 | 51.18 | 7.443 | 565.0 | 50.571 |
| | 317.8 | 43.64 | 29.622 | 608.9 | 10.730 |
| FCS-T | 320.3 | 44.13 | 29.391 | 606.2 | 10.898 |
| | 320.1 | 43.84 | 29.487 | 606.6 | 10.854 |
| stdev | 1.4 | 0.25 | 0.116 | 1.5 | 0.087 |
| average | 319.4 | 43.87 | 29.500 | 607.2 | 10.827 |

We have supplemented the test value of the two biomasses and statistical analysis label here.We modified the **Table 1** and **Table 2** in the manuscript as bellow:

**Table 1.** Basic properties of the soil in field experiments

| Soil | pH | Organic matter (g kg$^{-1}$) | Alkaline N (mg kg$^{-1}$) | Available P (mg kg$^{-1}$) | Available K (mg kg$^{-1}$) |
|---|---|---|---|---|---|
| Black soil | 6.55±0.31 | 51.18±1.41 | 7.44±0.57 | 565.0±2.3 | 59.00±0.85 |

Note: Values (± standard deviation) were averaged over 3 replicates.

**Table 2.** Elemental composition of materials used in field experiments

| Materials | C (g kg$^{-1}$) | H (g kg$^{-1}$) | N (g kg$^{-1}$) | O (g kg$^{-1}$) | C/N |
|---|---|---|---|---|---|
| CS | 376.4±1.0 | 51.18±0.33 | 7.44±0.03 | 565.0±0.8 | 50.57±0.08 |
| FCS-T | 319.4±1.4 | 43.87±0.25 | 29.50±0.12 | 607.2±1.5 | 10.83±0.09 |

Note: Values (± standard deviation) were averaged over 3 replicates. CS, corn straw; FCS-T, fermented corn straw treated with *T. reesei*.

Thank you very much for your consideration.
Kind regards,
(Yifeng Zhang)

First of all, we would like to thank the reviewers for their valuable comments and suggestions which will help us improve our manuscript. We have taken all of the reviewers' comments and suggestions when we were greatly revising our manuscript. Please find changes and explanations point by point below in Blue.

**RC3: 'Comment on soil-2021-105', Anonymous Referee #2, 08 May 2022**

**General comment**

➢ **Comment:** The paper of Zhang et al. deals with the effect of addition to soil of crop(corn) residues on the soil stable and labile organic C pools. The paper describes the results of a year-round field experiment where differently treated straw (FCS-T, CS) were added to so soil. I do not see great novelty in this work since the positive effect on SOM and SOM pools of the amendment with pre-treated crop residue is well known. The fermentation with *Tricoderma reesei* substantially mimed a composting effect, producing a material partly degraded and enriched of stable molecules. This fact, evidently favors both the SOM accumulation and production of labile substances from the portion not completely stabilized. This effect could be explored in future by testing different fermentation periods that should produce material with different stability.

The introduction is not very well informative about the study presented and, further, in some parts it has a textbook-style, which should be avoided in a research paper. The authors have to do a greater effort in displaying the state of the art in the recycling of crop residue as soil amendments and presenting their own hypotheses. I suggest to reduce the use of acronyms that make the paper very hard to follow. In my opinion the manuscript needs major revision before to be considered for publication.

*Response:* Thank you very much for your support of our manuscript. We have greatly revised our manuscript following the reviewers' comments and suggestion. We have deleted the textbook-style expressions, and reduced the use of acronyms for CPMI parameters, and keep the acronym of SOM to make the manuscript easier to understand and read.

Regarding the question of *novelty*, the information below tries to show and explain the relevance of the current research:

"Compared with direct application of crop residues and traditional composting methods, pretreatment of crop straws residues with microbial inoculants can effectively shorten the time for decomposition and humification of crop straws. This process fosters the increase in CPMI index, while also encouraging the accumulation of a more stable humus fraction of SOM. To-date, there are still conflicting reports and there is no general consensus on the effects of crop residue application on the formation process and composition of SOM.

The potential of using microbial-mediated fermented straw (especially *Trichoderma*-mediated straw fermentation) returned to the field as a crop residue management practice has gained renewed interest in recent years. This because when *Trichoderma*-mediated straw fermentation is used as a soil amendment and nutrient

source (Gaind and Nain. 2006; Gaind and Nain. 2007; Siddiquee et al., 2017) has great potential to increase crop yield (Islam et al., 2014), enhance plant development and reduce biotic and abiotic stresses (Sarangi et al., 2021). In previous studies on the application of fermented corn straw treated with *Trichoderma reesei* (*T. reesei*) in the field, Gaind and Nain. (2006), Gaind and Nain. (2007), and Sarangi et al. (2021) reported an increase in SOC and humus content in a series of studies. However, these studies did not go into considerable detail on the specific SOM components, such as the dynamic changes of labile organic carbon components, humic substance components, and carbon pool management level. We also observed that *T. reesei* has the strongest ability to form humic acid-like (Zhang et al., 2020; Zhang et al., 2021) during the decomposition of corn straw when compared with other fungi (*Phanerochaete chrysosporium* and *Trichoderma harzianum*). Although there are many studies, we still lack a clear understanding about the potential applications of *T. reesei* fermented corn straw and how it can increase SOM when incorporated into the field. Our findings presented the dynamics of this process in more detail, and we formulated hypothesis that suggest returning corn straw treated with *T. reesei* would be beneficial to increase labile SOC fractions and the humus composition. Our research hypothesis was validated by our results and we concluded that, amending fields with corn straw treated with *T. reesei* would be beneficial for the accrual of SOC. This is because the application of fermented corn straw treated by *T. reesei* was more advantageous in increasing the contents of aromatic C compounds (HA-C, and HM-C) and WEOC, which resulted in the overall increase in SOC and EOC, as well as the carbon pool management index. This also provides better evidence for the application of fermented corn straw treated by *T. reesei* in the field."

Regarding the reviewers' comment about: "*The fermentation with Tricoderma reesei substantially mimed a composting effect, producing a material partly degraded and enriched of stable molecules. This fact, evidently favors both the SOM accumulation and production of labile substances from the portion not completely stabilized. This effect could be explored in future by testing different fermentation periods that should produce material with different stability*."

We completely agree with the reviewer that the effect of fermentation with *Tricoderma reesei* should be explored in future by testing different fermentation periods that should produce material with different stability.

Indeed, in our parallel studies we are currently undertaking, we are researching about fermented corn straw treated with different microorganisms and under different humid and heat conditions, and fermentation time. The end materials will be returned to the field based on the humification effects produced.

We have significantly revised the Introduction section, reformulate a clearer research hypothesis, and put more effort into showcasing the latest technology in crop residue recycling as soil amendments. The revised Introduction section reads as follows (**page 1, line 23-page 2, line 84**):

[revised manuscript text omitted]

**Specific comments**

(page, line: comment)

➢ **Comment 1:** 2, 37-39: This sentence is a partial repetition of that at lines 25-28.

*Response:* We revised the sentence to read as follows (**page 2, line 46-47**):

Text: "Chen et al. (2017) and Ma et al. (2021) reported a significant increase in MBC and WEOC contents after crop straw residues were returned to the soil."

References: "Chen, Z., Wang, H., Liu, X., Zhao, X., Lu, D., Zhou, J., Li, C.: Changes in soil microbial community and organic carbon fractions under short-term straw return in a rice-wheat cropping system, Soil. Till. Res., 165(1), 121–127, doi:10.1016/j.still.2016.07.018, 2017.
Ma, L. J., Lv, X. B., Cao, N., Wang, Z., Zhou, Z. G., Meng, Y. L.: Alterations of soil labile organic carbon fractions and biological properties under different residue-management methods with equivalent carbon input, Appl. Soil. Ecol., 161, 103821, doi:10.1016/j.apsoil.2020.103821, 2021."

➢ **Comment 2:** 2, 40: Please, add how much time passed between the application of the straw and the determination of the C.

*Response:* Thank you for the suggestion. We added the time (which was five years) and we made revisions and updates in the text as follows (**page 2, line 47-49**):

Text: "In another study, Ndzelu et al. (2020b) also found that five years of corn straw application increased soil EOC, WEOC and MBC contents by 34.09%, 41.38% and 49.09% in the $0 - 20$ cm depth, respectively."

References: "Ndzelu, B. S., Dou, S., Zhang, X.: Corn straw return can increase labile soil organic carbon fractions and improve water-stable aggregates in Haplic Cambisol. J. Arid. Land., 12(6), 1018–1030, doi:10.1007/s40333-020-0024-7, 2020b."

➢ **Comment 3:** 2, 41-42: this sentence need references.

*Response:* We thank the reviewer, and the suggestion of the reviewer was adopted. We added some references in the text and reference list as follows (**page 2, line 49-5**1):

Text: "Therefore, assessing labile SOC fractions after crop straw applications may provide information about the formation of SOC (Chen et al., 2009; Huang et al., 2018; Liu et al., 2019; Ma et al., 2021)."

References: "Chen, H. Q., Hou, R. X., Gong, Y. S., Li, H. W., Fan, M. S., Kuzyakov, Y.: Effects of 11 years of conservation tillage on soil organic matter fractions in wheat monoculture in Loess Plateau of China, Soil. Till. Res., 106(1), 85－94, doi:10.1016/j.still.2009.09.009, 2009.
Huang, R., Tian, D., Liu, J., Lu, S., He, X., Gao, M.: Responses of soil carbon pool and soil aggregates associated organic carbon to straw and straw-derived biochar addition in a dryland

cropping mesocosm system, Agric. Ecosyst. Environ., 265, 576–586, doi:10.1016/j.agee.2018.07.013, 2018.

Liu, Z., Gao, T., Liu, W., Sun, K., Xin, Y., Liu, H., Wang, S., Li, G., Han, H., Li, Z., Ning, T.: Effects of part and whole straw returning on soil carbon sequestration in C3-C4 rotation cropland. J. Plant. Nutr. Soil Sci. 50, 73–85, doi:10.1002/jpln.201800573, 2019.

Ma, L. J., Lv, X. B., Cao, N., Wang, Z., Zhou, Z. G., Meng, Y. L.: Alterations of soil labile organic carbon fractions and biological properties under different residue-management methods with equivalent carbon input, Appl. Soil. Ecol., 161, 103821, doi:10.1016/j.apsoil.2020.103821, 2021."

➢ **Comment 4:** 2, 43: Please, explains what the CPMI index consists of.

*Response:* The suggestion of the reviewer was adopted. We have made revisions and updates in the text and references as below (**page 2, line 52-56**):

Text: "The carbon pool management index, an index that includes SOC pools (carbon pool index) and SOC lability (carbon pool activity index), is widely used as a sensitive tool to determine changes in soil C content (Blair et al. 1995; Duval et al. 2019). A high carbon pool management index indicates that soil management practices have a greater potential to promote soil C sequestration (Duval et al. 2019)."

References: "Blair, G., Lefroy, R., Lisle, L.: Soil carbon fractions based on their degree of oxidation, and the development of a carbon management index for agricultural systems, Aust. J. Agric. Res., 46, 1459–1466, doi:10.1071/AR9951459,1995.

Duval, M. E., Martinez, J. M., Galantini, J. A., Aitkenhead, M.: Assessing soil quality indices based on soil organic carbon fractions in different long-term wheat systems under semiarid conditions, Soil. Use. Manag., doi:10.1111/sum.12532 36:71－82, 2019."

➢ **Comment 5:** 2, 45-49. This part should be deleted. This information is well known by the soil community.

*Response:* The suggestion of the reviewer was adopted. We have deleted the part as suggested by the reviewer, and revised the opening paragraph in the manuscript to read as below (**page 2, line 57-59**).

Text: "Humic substance (HS) is the most stable fraction of SOM and contributes to the largest proportion to the total SOC (Olk et al., 2019; Dou et al., 2020). As a result, studying changes in soil humus components together with labile organic C fractions after corn straw application, could inform about the formation and stabilization of SOC during crop residues decay."

References: "Olk, D., Perdue, E., McKnight, D., Chen, Y., Farenhorst, A., Senesi, N., Chin, Y. P., Schmitt, K. P., Hertkorn, N., Harir, M.: Environmental and agricultural relevance of humic fractions extracted by alkali from soils and natural waters, J. Environ. Qual., 48(2): 217–232, doi:10.2134/jeq2019.02.0041, 2019.

Dou, S., Shan, J., Song, X., Cao, R., Wu, M., Li, C., Guan, S.: 2020. Are humic substances soil

microbial residues or unique synthesized compounds? A perspective on their distinctiveness, Pedosphere., 30(2), 159–167, doi:10.1016/s1002-0160(20)60001-7, 2020."

➢ **Comment 6:** 3, 76 The coordinate are not the same from line 69. Please explain why if is the case.

*Response:* Yes, the coordinates are different from where soil and corn straw residues were collected. Corn straw residues were collected from a corn field (125.394028, 43.812083; N43°48′43.5″, E125°23′38.50″) which was planted with corn in the previous season (i.e., 2018). Then in 2019, a year-round experiment was was conducted in a corn monocropping experimental field (125.402222, 43.818056; N43°49′5″, E125°24′8″) also located at Jilin Agricultural University, which was adjacent to where corn straw residues were collected.

We added that corn straw was collected from the adjacent field to clarify. Please see below revised sentence (**page 3, line 94-95**):

"Corn straw was collected from the adjacent cropland of corn (Zea mays L.) located at Jilin Agricultural University in Northeast China (N43°48′43.5″, E125°23′38.50″)."

➢ **Comment 7:** 3, 86: please define the mineral salt solution fertilization and the C/N ratio adjustment.

*Response:* Thank you to the reviewer for the suggestion. We have detailed the mineral salt solution that was mixed with the corn straw as follows (**page 4, line 106-110**):

Text: "The spore solution and a mineral salt solution (pH = 5) used were prepared similarly as described by Zhang et al (2020b), and the C/N ratio was adjusted to 25:1 using a mineral salt nutrient solution. The mineral salt nutrient solution (g $L^{-1}$) was prepared as a mixture of: $KH_2PO_4$ 28 g, $(NH_4)_2SO_4$ 9.6 g, $MgSO_4$ 4.2 g, $CoCl_2$ 4.2 g, $(NH_2)_2CO$ 2.2 g, $FeSO_4·7H_2O$ 0.07 g, $CaCl_2$ 0.028 g, $MnSO_4$ 0.021 g, $ZnSO_4$ 0.019 g, and the pH = 5. The fermentation process lasted 90 days and was carried out at 30 °C, 60% humidity, and 6.0 rpm."

References: "Zhang, Y., Dou, S., Hamza, B., Ye, S., Zhang, D.: Mechanisms of three fungal types on humic-like substances formation during solid-state fermentation of corn straw, Intl. J. Agric. Biol., 24, 970–976, doi:10.17957/IJAB/15.1377, 2020."

➢ **Comment 8:** 4, 114-116. how long lasted the shaking time with water?

*Response:* The shaking time with water lasted for 60 minutes. The sentence was revised to read as below (**page 4, line 117-120**):

Text: "The WEOC content was obtained by successively extracting 5 g of air-dried soil samples with distilled water in a 1:6 ratio of soil to water. The soil-solution mixture was shaken on a reciprocal shaker at 25 ∘C for 60 min, and then centrifuged at 4500 rpm for 20 min. The solution was filtered through a 0.45-μm filter membrane (Changtingny et al., 2010)."

References: "Changtingny, M. H., Curtin, D., Beare, M. H., Greenfield, L. G.: Influence of Temperature on Water-Extractable Organic Matter and Ammonium Production in Mineral Soils, Soil. Sci. Soc. Am. J., 74(2), 517–524, doi:10.2136/sssaj2008.0347, 2010."

➢ **Comment 9:** 4, 117: No incubation and fumigation before the MBC extraction? How did you estimate the MBC without the difference between fumigated and not-fumigated samples? The simple extraction with $K_2SO_4$ gives you the OM soluble in a $K_2SO_4$ solution which has no connection with the microbial C!!

*Response:* Thank you to the reviewer for raising this issue. Indeed, in our previous version of the manuscript we did not detail the extraction procedure of MBC, we simply cited the reference we followed for the extraction procedure. But, upon the reviewers' comment, we added details of MBC estimation to provide some clarity. This was added as follows (**page 5, line 141-149**):

Text: "Fresh soil equivalent to 10 g of oven-dried soil was fumigated with $CHCl_3$ for 24 h and the other 10 g of soil was not fumigated. Both fumigated and unfumigated soils were then extracted with 0.5 mol $L^{-1}$ $K_2SO_4$. The MBC content was estimated from the increase in organic C in the 0.5 mol $L^{-1}$ $K_2SO_4$ extracts of $CHCl_3$ fumigated soils as described by Vance et al. (1987). The soil WEOC and MBC contents were determined by a TOC analyser (Shimadzu TOC-VCPH, Japan). The soil WEOC and MBC contents were determined by a TOC analyser (Shimadzu TOC-VCPH, Japan). MBC was calculated as below Eq. (1):

$$MBC = \frac{F_c}{k_c} \tag{1}$$

where $F_c$ is the difference between the amount of $CO_2$ released by fumigated and unfumigated soil (control) during the cultivation period; $k_c$ is the conversion coefficient. "

References: "Vance, E. D., Brookes, P. C., Jenkinson, D. S.: An extraction method for measuring soil microbial biomass C. Soil. Biol. Biochem., 19(6), 703–707, doi:10.1016/0038-0717(87)90052-6, 1987."

➢ **Comment 10:** 5, 120-122: Here the authors refer to fumigation, but it is not clear. Please xplain better the methodology used. When and how did you fumigate soil during the cultivation period?

*Response:* Thank you to the reviewer for the suggestion. We collected fresh soil samples at 0 d, 30 d, 60 d, 90 d, 180 d, 360 d during the cultivation period. In each soil sampling day, the collected fresh soil was picked out all visible corn straw materials and passed through a 2 mm sieve. Then the soil sample was transported to the lab to analyze MBC in soil. The description of soil sampling in **2.3.2 Soil sampling and analysis (page 5, line 130-134)**. During each sampling time, MBC was estimated according to Vance et al. (1987), as briefly described in the *Reponse* of

**comment 9**.

Text: "**2.3.2 Soil sampling and analysis**

Five topsoil samples (0 – 20 cm) were collected from each plot at 0 d, 30 d, 60 d, 90 d, 180 d, and 360 d using a stainless-steel soil auger (5 cm in diameter). For each soil sampling day, all visible corn straw materials in CS and FCS-T soils were picked out with tweezers and returned to their respective plots. The collected fresh soil was immediately divided into two sub-samples and passed through a 2 mm sieve. One subsample was then placed in a refrigerator (4 ∘C) to later analyze MBC in soil. The remaining subsample was air-dried to determine SOC, EOC, WEOC content and humus composition."

"Fresh soil equivalent to 10 g of oven-dried soil was fumigated with $CHCl_3$ for 24 h and the other 10 g of soil was not fumigated. Both fumigated and unfumigated soils were then extracted with 0.5 mol $L^{-1}$ $K_2SO_4$. The MBC content was estimated from the increase in organic C in the 0.5 mol $L^{-1}$ $K_2SO_4$ extracts of $CHCl_3$ fumigated soils as described by Vance et al. (1987). The soil WEOC and MBC contents were determined by a TOC analyser (Shimadzu TOC-VCPH, Japan). The soil WEOC and MBC contents were determined by a TOC analyser (Shimadzu TOC-VCPH, Japan). MBC was calculated as below Eq. (1):

$$MBC = \frac{F_c}{k_c} \tag{1}$$

where $F_c$ is the difference between the amount of $CO_2$ released by fumigated and unfumigated soil (control) during the cultivation period; $k_c$ is the conversion coefficient. "

References: "Vance, E. D., Brookes, P. C., Jenkinson, D. S.: An extraction method for measuring soil microbial biomass C. Soil. Biol. Biochem., 19(6), 703 – 707, doi:10.1016/0038-0717(87)90052-6, 1987."

➢ **Comment 11:** 5, 138: delete CPMI = CPI â x CPAI x 100
*Response:* Thank you for the suggestion. We deleted the repeated CPAI equation as advised by the reviewer (**page 6, line 166**).

➢ **Comment 12:** 5, 146-147: The humification degree (PQ) was calculated as HA-C/HE-C ratio (Sugahara and Inoko, 1981).
*Response:* Thank you for the suggestion, we re-wrote the sentence following the suggestion of the reviewer (**page 6, line 174-175**).

Text: "The humification degree (PQ) was calculated as HA-C/HE-C ratio (Sugahara and Inoko, 1981)".

➢ **Comment 13:** 8, 220, 221: Barley.

*Response:* We thank the reviewer for picking up this error. We change "barly" to "barley" throughout the discussion where the mistake appeared (**page 9, line 258-260**).

Text: "Ma et al. (2021) reported similar findings with barley treated with microbial inoculant, in which the WEOC content was significantly higher than that of barley residue without microbial inoculant, but the EOC content differed seldomly."

➢ **Comment 14:** 8, 239-250. This part describe essentially the effect of addition to soil of pretreated material (e.g. compost). I think that this part does not add very much to the discussion and could be diluted along the section.

*Response:* Thank you to the reviewer for the suggestion. We greatly revised this part, by deleting most it and handful of lines were merged with section 4.1. Now the section reads as follows (**Page 8 Line 232-Page 10 Line 286**):

[revised manuscript text omitted]

Thank you very much for your consideration.
Kind regards,
(Yifeng Zhang)

---

## Author Response (AR2)

**Soil**
**Manuscript No.:** SOIL-2021-105
**Manuscript Title:** Effects of returning corn straw and fermented corn straw to fields on the soil organic carbon pools and humus composition
**Article Type:** Research paper

**Authors:** Yifeng Zhang, Sen Dou, Batande Sinovuyo Ndzelu, Rui Ma, Dandan Zhang, Xiaowei Zhang, Shufen Ye, Hongrui Wang

**Topical Editor Comments to the author**

Comments to the author:

Dear Authors,

the revised manuscript was very improved but still needs minor revisions before publication.

English editing is strongly required especially in the new sentences added within the revision.

The second point is that Introduction should be still reduced to the most important points that drive the reader to understand the state of the art and the main gaps that lead to formulate the hypotheses and the objectives of the work.

Conclusions remain a summary of results rather than highlighting the conclusive remarks obtained by the current findings.

Thus I ask the Authors to make these changes before proceeding to publication.

Best regards,
Luisella Celii

**Response to Topical Editor comments**

Dear Editor Luisella Celi,

First of all, on behalf of all co-authors, we appreciate your positive and constructive comments and suggestions on our manuscript submitted to SOIL. You have given us invaluable advice, from the preliminary review stage, the interactive discussion stage, and the revision stage.

The English of the manuscript was thoroughly revised by one of the co-author (Batande Sinovuyo Ndzelu) a native English speaker. We revised the entire manuscript with the special attention paid at the newly added sentences.

The Topical Editor comments are laid out below and specific concerns have been numbered. Please find changes and explanations point by point below in **Blue**. All changes in the revised manuscript are made using Track Changes to make reviewing easy. We believe that our new revised version of the manuscript has been much improved.

Yours sincerely,
Dr. Yifeng Zhang

**Topical Editor specific comments**

(page, line: comment)

➢ **Comment 1:** English editing is strongly required especially in the new sentences added within the revision.

*Response:* We have asked that a native English co-author (Batande Sinovuyo Ndzelu) to thoroughly revise the English of the manuscript, particularly the newly added sentences. And all English editing changes in the revised manuscript are made using Track Changes in blue to make reviewing easy.

➢ **Comment 2:** The second point is that Introduction should be still reduced to the most important points that drive the reader to understand the state of the art and the main gaps that lead to formulate the hypotheses and the objectives of the work.

*Response:* Thank you for the suggestion. We have reduced the Introduction to the most important points that shows the state of the art and main points we are trying to show the reader. Consequently, the introduction part of the manuscript reduced from 1118 to 937 words, while not changing important parts of the original revision. We have made revisions and updates in the text as follows (**page 1, line 19-page 3, line 69**):

[revised manuscript text omitted]

> **Comment 3:** Conclusions remain a summary of results rather than highlighting the conclusive remarks obtained by the current findings.

*Response:* Thank you for the suggestion. Following the comments of the editor, we rewrote the conclusions and highlighted the main outcomes and current findings of the study. Now the Conclusions reads as below (**page 10, line 283-301**):

Text: "**5 Conclusion**

In this 360-day field experiment, we applied corn straw (CS) and fermented corn straw treated with *Trichoderma reesei* (FCS-T) under equal C input, and a blank control treatment (CK) for comparison. The following conclusions were drawn:

The change of SOC content mainly depends on the C content of the stable soil components, i.e., aromatic compounds (HA-C and HM-C). The FCS-T material has a lower C/N ratio, higher alkyl and aromatic C content, and humic-like substances. When the FCS-T material is applied to the soil, it is more advantageous in promoting the soil humification process and increasing soil HA-C and HM-C content. Compared with direct corn straw application (i.e., CS treatment), the FCS-T treatment increased the SOC content by 1.715 g kg$^{-1}$ on the 360$^{th}$ day, and increased the PQ value to 74.1%. The application of FCS-T material with a lower C/N ratio sequestered more SOC than the application of CS, which supported the idea that the C/N ratio in the organic amendments is negatively correlated with SOC content.

The application of FCS-T significantly improves the release of soil labile carbon components. In particular, the WEOC content could maintain a high level for a long time, while the EOC and MBC contents of FCS-T treatment increased significantly more than CS treatment on the 60$^{th}$ and 90$^{th}$ days.

Compared to CS treatment, the FCS-T treatment significantly improved the level of carbon pool management to 13.95, primarily by promoting the simultaneous increase in the contents of EOC and stable organic carbon components (HA-C and HM-C).

The results confirmed our initial hypothesis that the application of FCS-T has a greater potential to increase soil carbon sequestration compared with direct application of CS. As a method of returning straw residues to the field, the application of FCS-T is a practice worthy of further exploration."

**Adjustments other than Topic editor comments**

➢ The clarification regarding the changes in co-author affiliations

*Response:* Although the two co-authors (Batande Sinovuyo Ndzelu and Xiaowei Zhang are now attending new institutions, they were at Jilin Agricultural University when they participated in this study. So all the co-authors jointly decided not to show affiliations b and c.

➢ The clarification regarding the changes of Figure 7

*Response:* The Figure 7 of the previous version of the manuscript was too small to read. We have therefore changed the Figure 7 to a more readable size without making changes to the data its self. we have only improved the quality of Figure 7 to make it easier to read, and the new version and past version are as follows:

New version:

[Figure]

Past version:

[Figure]

Thank you very much for your consideration.
Kind regards,
(Yifeng Zhang)